# RetinexGAN Enables More Robust Low-Light Image Enhancement Via Retinex Decomposition Based Unsupervised Illumination Brightening

## Abstract

Most existing image enhancement techniques rely heavily on strict supervision of paired images. Moreover, unsupervised enhancement methods also face challenges in achieving a balance between model performance and efficiency when handling real-world low-light images in unknown complex scenarios. Herein, we present a novel low-light image enhancement scheme termed **RetinexGAN** that can leverage the supervision of a limited number of low-light/normal image pairs to realize an accurate Retinex decomposition, and based on this, achieve brightening the illumination of unpaired images to reduce dependence on paired datasets and improve generalization ability. The decomposition network is learned with some newly established constraints for complete decoupling between reflectance and illumination. For the first time, we introduce the feature pyramid network (FPN) to adjust the illumination maps of other low-light images without any supervision. Under this flexible framework, a wide range of backbones can be employed to work with illumination map generator, to navigate the balance between performance and efficiency. In addition, a novel attention mechanism is integrated into the FPN for giving the adaptability towards application scenes with different environment like underwater image enhancement (UIE) and dark face detection. Extensive experiments demonstrate that our proposed scheme has a more robust performance with high efficiency facing various images from different low-light environments over state-of-the-art methods.

## 1 Introduction

Image enhancement is a vital data preprocessing technology which can improve the image quality and strengthen the image interpretation and recognition performance. It has multifarious applications ranging from underwater target detection Islam et al. (2018); Yu et al. (2020); Bernardi et al. (2022), low light face recognition Wang et al. (2008) and scene understanding Dvornik et al. (2019); Mottaghi et al. (2015); Nauata et al. (2019), etc. However, those application scenarios are all vulnerable to external lighting conditions and they also have many open challenges. Hence, image enhancement is a necessary data preprocessing means to ameliorate the visual quality of the images directly received from vision sensor so as to realize a significant improvement the capability of visual information perception.

However, there are many problems in existing image enhancement researches. For traditional methods like Land & McCann (1971); Jang et al. (2012); Li et al. (2018); Gilboa et al. (2004); Bettahar et al. (2011), it will be more time-consuming to process each image, so there is no way to achieve the enhancement for large-scale images in batches quickly which further limits the potential application of those traditional enhancement algorithms in many popular fields. The supervised learning based methods like Lore et al. (2017); Chen Wei (2018); Zhang et al. (2021); Moran et al. (2020) alleviates the problem of timeliness in traditional methods to some extent. Yet they need sufficient number of paired images to learning the mapping rules and they are usually trained by synthetic corrupted images which are usually not photo-realistic enough, leading to various artifacts when the trained model is applied to real-world low-light images in different environment. As for the existing

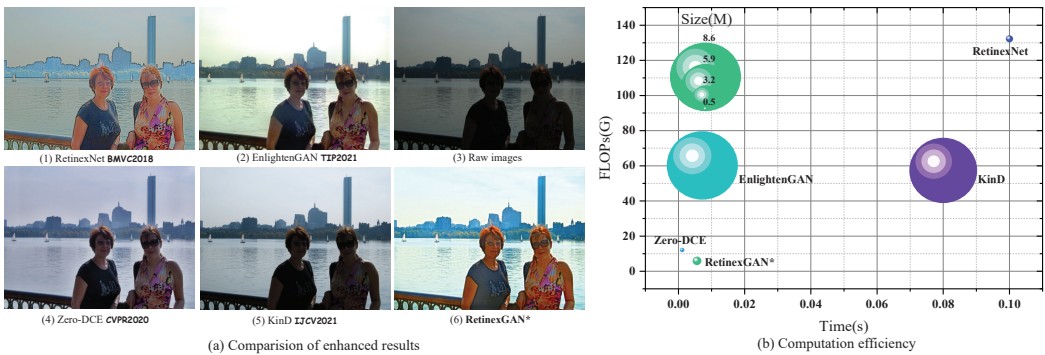

(a) Comparision of enhanced results  (b) Computation efficiency

Figure 1: This figure shows the enhanced version of low light image from different algorithms and their computation efficiency. It indicate that the balance between model performance and efficiency is hard to navigate for those existing works.

unsupervised methods Jiang et al. (2021); Li et al. (2021); Ma et al. (2022); Liu et al. (2021), they all suffer from the difficulty in balancing between model performance and efficiency when handling real-world low-light images in wild environment. We show some enhancement examples of existing state-of-the-art methods in Figure 1. More discussion on related work is provided in the Appendix.

To address the aforementioned challenges, we propose a novel semi-supervised illumination brightening framework, referred to as RetinexGAN, which is designed on the basis of Retinex decomposition. This approach aims to achieve a well-balanced performance and efficiency, while also offering flexibility in low-light image enhancement by utilizing lightweight CNNs for data-driven Retinex decomposition and FPN with different kinds of pretrained backbones for down-sampling in illumination generation. In conclusion, the contributions of this paper are three-fold. **(1)** We design a lightweight CNN model to realize the data-driven Retinex decomposition. The decomposition model is learned with some vital constraints including the consistent reflectance and consistent contrast shared by paired low/normal-light images, domain translation constancy and feature constancy of reflectance in high dimension mapped by VGG19. **(2)** We, for the first time, present a novel utilization of the FPN with a diverse array of backbones and unsupervised training loss with spatial attention mechanisms is defined to brighten the illumination maps, endowing the adaptation capability towards diverse application scenes with different exposure levels. **(3)** We undertake comprehensive experiments to elucidate the superiority of our approach in comparison to other state-of-the-art methods. Our evaluation encompasses diverse image datasets, encompassing both synthetic and real world low light images, serving as test samples to corroborate the robustness coming up against various images in different scene and capacity of our method to maintain a balance between performance and efficiency across various low-light scenarios.

## 2 METHODOLOGY

In this section, we first introduce the data-driven Retinex decomposition module with novel learning constraints which can provide prior information for lateral illumination adjustment. Then we continue to describe the FPN based illumination generator with novel spatial attention mechanism and its non-reference loss function is also presented.

### 2.1 NETWORK ARCHITECTURE

**Deep Decomposition Network:** Detailed information about deep decomposition network in this paper please refer to Appendix.

**Illumination Generator:** We use the pretrained backbones like Inception-ResNet-v2 and MobileNet to complete down sampling. As shown in Fig.2, the size of the five feature maps are set to $\frac{1}{2}$, $\frac{1}{4}$, $\frac{1}{8}$, $\frac{1}{16}$ and $\frac{1}{32}$ of the input image. The feature maps with $\frac{1}{4}$, $\frac{1}{8}$, $\frac{1}{16}$ and $\frac{1}{32}$ size of the input image are then concatenated into one tensor to reconstruct the parameter maps in higher spatial resolution. Moreover, we use a Sigmoid activation function to constrain the output of the model.

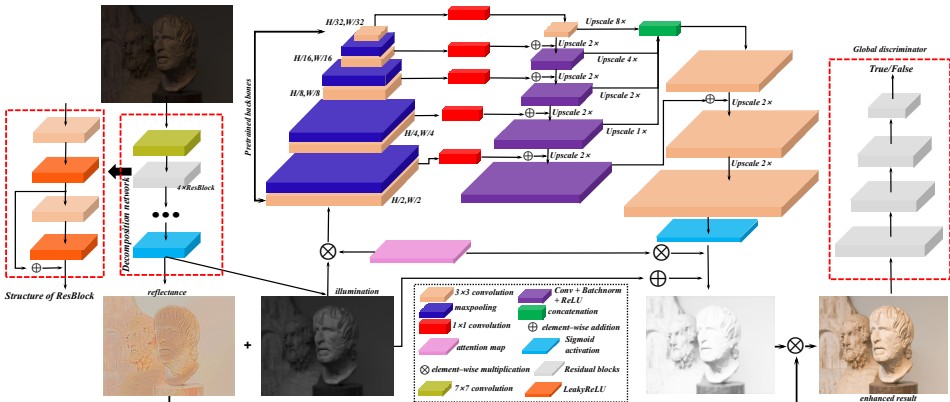

Figure 2: The framework of RetinexGAN. The RetinexGAN contains a deep decomposition network, an illumination generator (discriminator for training). The deep decomposition network is designed to realize the data-driven Retinex decomposition and the illumination generator is established for adjusting the illumination maps of low-light images that are utilized to synthesize the final enhanced results.

To effectively address potential issues related to overexposure in the generated illumination maps, we have integrated spatial attention mechanisms post the output layer. Moreover, to enhance the model's generalization capacity, we have introduced global residual connections into the architecture. These methodological enhancements serve to bolster the overall performance and robustness of the model.

**Global Discriminator:** Detailed information about discriminator in this paper please refer to Appendix.

## 2.2 Data-Driven Retinex Decomposition

Our method is founded upon the Retinex model, which elucidates the constitution of a image:

$$\mathbf{I}_{x/y} = \mathbf{R}_{x/y} \otimes \mathbf{L}_{x/y} \tag{1}$$

where $\mathbf{I}_x$, $\mathbf{I}_y$ represent the images in the underexposure domain and normal-exposure domain. $\mathbf{R}_x$, $\mathbf{R}_y$ denote the reflectance of $\mathbf{I}_x$ and $\mathbf{I}_y$. $\mathbf{L}_x$, $\mathbf{L}_y$ denote the illumination of $\mathbf{I}_x$ and $\mathbf{I}_y$. And $\otimes$ stand for the element-wise multiplication.

The data-driven Retinex decomposition is realized by learning the potential mapping rules through paired images. To address this issue, we have devised a set of carefully crafted constraints to effectively steer the optimization trajectory of the model.

**Reconstruction Constraints:** In the context of the traditional Retinex theory, it is postulated that any given image can be effectively disentangled into two distinct components, namely the reflectance and the corresponding illumination maps. This fundamental property allows for the reconstruction of the original image, irrespective of whether it is underexposed or not. Thus we can formulate the most basic constraints as follows.

$$\mathcal{L}_r = ||\mathbf{R}_x \otimes \mathbf{L}_x - \mathbf{I}_x||_1 + ||\mathbf{R}_y \otimes \mathbf{L}_y - \mathbf{I}_y||_1 \tag{2}$$

**Domain Translation Constraints:** As per the traditional Retinex theory, the domain translation of images can be accomplished by exchanging illumination maps. In our research, we aim to ensure that the decomposition results retain this intrinsic attribute throughout our proposed approach. It corresponds roughly to the self-reconstruction loss which can be expressed as follows.

$$\mathcal{L}_{do} = ||\mathbf{R}_x \otimes \mathbf{L}_y - \mathbf{I}_y||_1 + ||\mathbf{R}_y \otimes \mathbf{L}_x - \mathbf{I}_x||_1 \tag{3}$$

**Reflectance Feature Constancy Constraints:** We hope to ensure consistency between the reflectance of low light images and their corresponding images in the normal-exposure domain, while

simultaneously preserving consistency with the features of their respective source images in high-dimensional space. Thus we design the reflectance feature constancy constraints expressed as follows.

$$\mathcal{L}_f = ||\mathbf{R}_x - \mathbf{R}_y||_1 + \sum_{i \in \mathcal{C}} \lambda_i \left( ||\phi_i(\mathbf{R}_x) - \phi_i(\mathbf{I}_x)||_1 + ||\phi_i(\mathbf{R}_y) - \phi_i(\mathbf{I}_y)||_1 \right) \qquad (4)$$

where $\phi_i$ denotes the feature maps from the $i$-th feature layer in the VGG19 pretrained on ImageNet. $\mathcal{C}$ represents the set of the index of feature layer that can be formulated as $\mathcal{C} = \{2, 7, 12, 21, 30\}$.

**Contrast Consistency Constraints:** The contrast ratios of reflectance maps are also essential issues to improve their visual quality. We design this loss term to encourage the reflectance maps to maintain the same contrast as images in the normal-exposure domain. It can be represented as follows.

$$\mathcal{L}_c = \sum_{i=1}^{N_p} \sum_{j \in \Omega_i} \frac{\left(\mathbf{R}_y^i - \mathbf{R}_y^j\right)^2}{N_p} - \sum_{i=1}^{N_p} \sum_{j \in \Omega_i} \frac{\left(\mathbf{R}_x^i - \mathbf{R}_x^j\right)^2}{N_p} \qquad (5)$$

where $N_p$ represent the number of pixels in the image. $\Omega_i$ denote the set of eight pixels around pixel $i$ (shown in Supplementary materials). $\mathbf{R}_y^i$ and $\mathbf{R}_x^i$ represent the pixels of the reflectance maps obtained from images in the normal-exposure domain and images in the underexposure domain.

**Total Variation Constraints:** In our work, we further modified the weighted TV loss to refine the demand for gradients in the strong edge parts of the illumination maps to prevent the generation of black edges in the corresponding reflectance maps. It can be expressed as follows.

$$\mathcal{L}_{tv} = |\nabla \mathbf{L}_y + \nabla \mathbf{L}_x| \exp\left(-\lambda_s \max\left(\nabla \mathbf{R}_x, \nabla \mathbf{R}_y\right)\right) \qquad (6)$$

Accordingly, we can derive the total constraints as follows.

$$\mathcal{L} = \mathcal{W}_r \mathcal{L}_r + \mathcal{W}_{do} \mathcal{L}_{do} + \mathcal{W}_c \mathcal{L}_c + \mathcal{W}_f \mathcal{L}_f + \mathcal{W}_{tv} \mathcal{L}_{tv} \qquad (7)$$

where $\mathcal{W}_r$, $\mathcal{W}_{do}$, $\mathcal{W}_c$, $\mathcal{W}_f$ and $\mathcal{W}_{tv}$ are weights of those constraints.

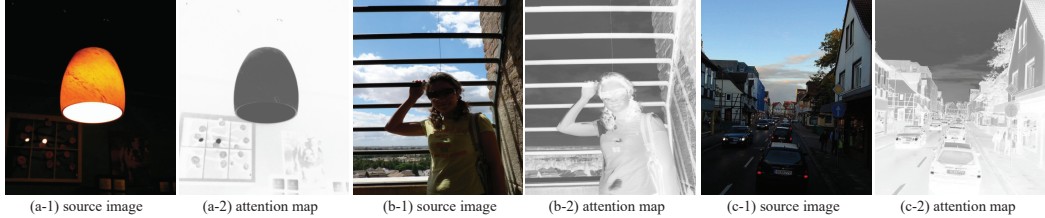

| (a-1) source image | (a-2) attention map | (b-1) source image | (b-2) attention map | (c-1) source image | (c-2) attention map |

Figure 3: We have shown three groups of low-light images and their attention maps. The spatial attention mechanism assigns lower attention values to the brighter regions in the source image, while assigning higher attention values to the darker regions.

## 2.3 ILLUMINATION BRIGHTENING WITHOUT SUPERVISION

The spatial attention mechanism and non-reference losses proposed by us play a pivotal role in achieving unsupervised optimization for illumination adjustment. Herein, we will delve into the detailed exposition of these two components.

### 2.3.1 SPATIAL ATTENTION MECHANISM

As shown in Fig.2, we have designed a attention map to suppress local overexposure and local underexposure in the final enhanced results. First, the low-light image $\mathbf{I}_x$ is transformed into Hue, Saturation, Value (HSV) space and we let $\mathbf{V}_{\mathbf{I}_x}$ define the value channel of $\mathbf{I}_x$ in HSV space. The attention map can be designed as follows.

$$\mathcal{A}_x = \exp\left(-2\mathbf{V}_{\mathbf{I}_x}\right) \qquad (8)$$

Some illustrations of our proposed spatial attention mechanism have been shown in Fig.3. This design allows our proposed model to prioritize attention on the darker regions of the image, while making minimal adjustments to areas with normal brightness. This aspect holds significant importance, particularly for low-light images exhibiting uneven illumination distributions. And more importantly, our proposed spatial attention mechanism can endow our model with the capability to adapt to diverse scenes with different exposure levels as mentioned above.

### 2.3.2 NON-REFERENCE LOSS FUNCTION

Since this learning process don't have any supervision of paired images, we need to design a differentiable non-reference loss function to evaluate the quality of final enhanced results.

**Brightness Control Loss:** The average pixel value of each region can reflect the its illumination intensity. We use the average pooling operation to calculate the mean value of a specific region in the illumination map. The kernel size of the average pooling is denoted as $K_b$. The loss term can be formulated as follows.

$$B_{\mathrm{mean}} = \mathrm{AvgPooling}_{K_b}\left(\mathbf{L}_x\right), \ \mathcal{J}_b = \|B_{\mathrm{mean}} - B_r\|_2 \tag{9}$$

where $B_r \in (0,1)$ represent a reference average pixel value.

**Feature Retention Loss:** We hope the generator to produce an illumination map that preserves the fine details present in the illumination map of the source image. Thus the loss term can be expressed as follows.

$$\mathcal{J}_f = \sum_{i \in \mathcal{C}} \gamma_i ||\phi_i\left(\hat{\mathbf{L}}_x\right) - \phi_i\left(\mathbf{L}_x\right)||_1 \tag{10}$$

where $\hat{\mathbf{L}}_x$ represent the output of illumination generator. $\phi_i$ denotes the feature maps from the $i$-th feature layer in the VGG19 pretrained on ImageNet. And the setting of $\mathcal{C}$ is the same as that in Eq.(4).

**Attention Loss:** To mitigate the occurrence of overexposure and underexposure, our objective is to allocate higher pixel increments $\Delta\mathbf{L}_x = \hat{\mathbf{L}}_x - \mathbf{L}_x$ to low-intensity regions, while applying lower increments to high-intensity regions. This approach ensures a balanced enhancement of image brightness across varying illumination levels. We use cosine similarity to express this loss term as follows.

$$\mathcal{J}_a = \left(1 - \frac{\left(\mathrm{vec}\left(\Delta\mathbf{L}_x\right)\right)^{\mathrm{T}} \mathrm{vec}\left(\boldsymbol{\mathcal{A}}_x\right)}{\|\mathrm{vec}\left(\Delta\mathbf{L}_x\right)\|_2 \|\mathrm{vec}\left(\boldsymbol{\mathcal{A}}_x\right)\|_2}\right) \tag{11}$$

**Adversarial Loss:** This loss term is purposefully formulated to guide the generator towards generating illumination maps to synthesize a fake image that exhibits a similar level of realism compared to real images. It can be formulated as follows.

$$\mathcal{J}_{adv} = \left(D\left(\mathbf{I}_f\right) - 1\right)^2 \tag{12}$$

where $D$ denotes the forward function of the global discriminator. $\mathbf{I}_f = \mathbf{R}_x \otimes \hat{\mathbf{L}}_x$ represents the synthetic images.

Finally, the total loss function can be expressed as:

$$\mathcal{J} = \mathcal{O}_b\mathcal{J}_b + \mathcal{O}_f\mathcal{J}_f + \mathcal{O}_a\mathcal{J}_a + \mathcal{O}_{adv}\mathcal{J}_{adv} + \mathcal{L} \tag{13}$$

where $\mathcal{O}_b$, $\mathcal{O}_f$, $\mathcal{O}_a$ and $\mathcal{O}_{adv}$ are positive constant which serve as the weights of the loss terms.

## 3 EXPERIMENTS

In this section, we will introduce the experimental results of our proposed RetinexGAN framework. In Section 3.1, the implementation details including the structural setting, learning parameters setting of our proposed method are elaborated to make our algorithms easier to be followed. Then in Section 3.2 we carry on an ablation study on the RetinexGAN. Third in 3.3, benchmark evaluation of the entire proposed enhancement framework is shown to the readers which involves qualitative experiments through visual and perceptual comparisons and quantitative experiments through quality evaluation metrics comparison.

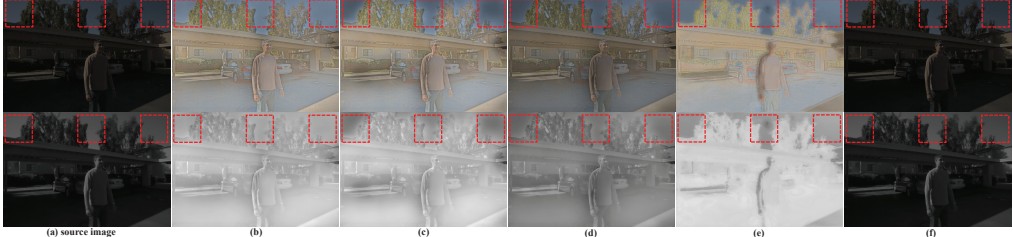

Figure 4: Visual comparison from the ablation study (contribution of losses) of RetinexGAN. Row 1 (Column (b)-Column (f)) display the final enhanced results. Row 2 (Column (b)-Column (f)) are the adjusted illumination maps generated by illumination generator. Column (a) shows the source image (Row 1) and its illumination maps (Row 2). (b) RetinexGAN. (c) $w/o$ $\mathcal{J}_a$. (d) $w/o$ $\mathcal{J}_{adv}$. (e) $w/o$ $\mathcal{J}_f$. (f) $w/o$ $\mathcal{J}_b$

## 3.1 IMPLEMENTATION DETAILS

The training images in LOL dataset Chen Wei (2018) is employed to train the deep decomposition network for learning the mapping rules of Retinex decomposition. Besides, we employ 500 low-light images and 500 normal-exposure images from the SCIE dataset (Part 1) Cai et al. (2018) to train the unsupervised illumination generator model. All those training images are resized to the size of $480 \times 640 \times 3$.

We implement our framework with Pytorch on one NVIDIA RTX 3090 GPUs. The batch size in training is set to be 2. The kernel weights and bias of each layer in the models (except for the down-sampling in FPN) are initialized with standard zero mean and $0.1$ standard deviation. Downsampling operations in FPN are initialized by different pretrained backbones (e.g., Inception-ResNet-v2 and **MobileNet**). We freeze their parameters in the first 5 training steps. The optimizer used in our framework are all Adam optimizers with default parameters and fixed learning rate 0.0001. The weight parameters are set to be $\mathcal{W}_r = 1$, $\mathcal{W}_{do} = 0.001$, $\mathcal{W}_c = 0.1$, $\mathcal{W}_f = 0.2$, $\mathcal{W}_{tv} = 0.1$, $\mathcal{O}_c = 50$, $\mathcal{O}_b = 15$, $\mathcal{O}_{tv} = 150$, $\mathcal{O}_{col} = 5$. The pseudo codes of training process is provided in the Appendix.

## 3.2 ABLATION STUDY

**Contribution of Each Loss.** We show the visualization results of the contribution of each loss in ablation study in Figure 4 and Figure 9 (see it in Appendix). It can be seen that the brightness of some regions (red box marking) in the results without $\mathcal{J}_a$ has not been fully improved. This shows the importance of attention loss in enhancing images with uneven brightness distribution. The adversarial loss $\mathcal{J}_{adv}$ is designed to make the synthesized image more realistic. The results without $\mathcal{J}_{adv}$ have lower visual quality than the full results. Severe loss of texture details in the adjusted illumination maps occurs in the results without $\mathcal{J}_f$. Finally, removing the brightness control loss $\mathcal{J}_b$ hampers the restoration of low light regions to normal exposure levels.

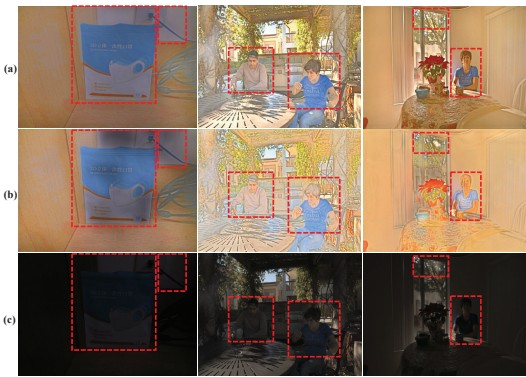

Figure 5: Visual comparison from the ablation study (contribution of attention map) of Retinex-GAN. (a): RetinexGAN. (b): $w/o$ $\mathcal{A}_x$. (c): Input. Please zoom in view to see the details.

**Contribution of Attention Map.** We evaluate the impact of our designed spatial attention mechanism. The visual comparison is given in Figure 5. Images in Row (b) suffer from severe color distortion and overexposure/underexposure issues. Particularly in column 3, it is observed that the results obtained from RetinexGAN without

$\mathcal{A}_x$ exhibit both color distortion and overexposure, leading to the loss of fine texture details in the image.

**Selection of Pretrained Backbones.** The pretrained backbones in the FPN based illumination generator are essential to adjust the illumination maps of the low light images. We present the results of the impact on the selection of pretrained backbones in Table 4. Although the Inception-ResNet-v2 backbone can obtain enhanced images with better quality evaluation metrics, it has a large size of trainable parameters and FLOPs. On the other hand, the results from **MobileNet** have the second best evaluation metrics. It is of significance to note that while the enhanced images achieved by **MobileNet** exhibit a minor disparity in comparison to those yielded by Inception-ResNet-v2, the **MobileNet** notably presents a substantial reduction in computational load and inference time in contrast to the Inception-ResNet-v2. Both of the DenseNet and SENet backbones have large number of parameters and high computational complexity, yet their performance is weaker than the **MobileNet** and Inception-ResNet-v2. Hence, the adoption of the **MobileNet** backbone facilitates our proposed RetinexGAN in attaining a commendable balance between performance and efficiency.

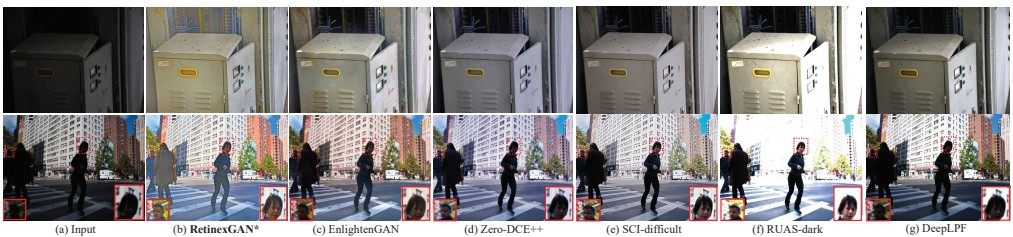

Figure 6: Visual comparison from the benchmark evaluation (LSRW dataset, GladNet dataset). We have zoomed in view of some regions of interest to analyze the enhanced results in details.

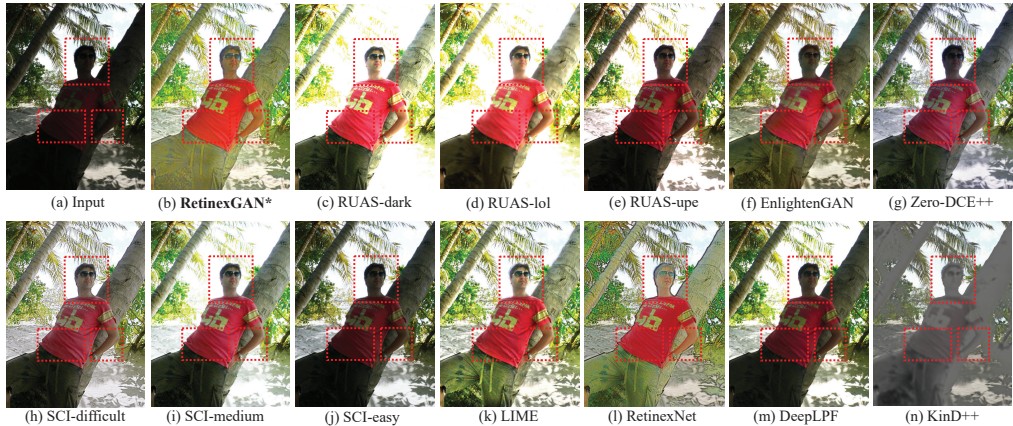

Figure 7: Visual comparison from the benchmark evaluation (VV dataset).

## 3.3 BENCHMARK EVALUATIONS

We use many challenging low light image from LSRW Hai et al. (2023), NPE Wang et al. (2013), MEF Ma et al. (2015), VV[1], GladNet, ExDark Loh & Chan (2019), HDR Kalantari & Ramamoorthi (2017) dataset in this subsection. The algorithms participating in the benchmark evaluation can be divided into traditional method (LIME Guo et al. (2016)), supervised learning (DeepLPF Moran et al. (2020), RetinexNet Chen Wei (2018), KinD++ Zhang et al. (2021)) and unsupervised learning (EnlightenGANJiang et al. (2021), Zero-DCE++ Li et al. (2021), RUAS Liu et al. (2021), SCI Ma et al. (2022)). The comparative experiments are mainly conducted through qualitative analysis of visual perception, quantitative analysis of evaluation metrics and face detection in dark.

---

[1]https://sites.google.com/site/vonikakis/datasets

**Visual and Qualitative Comparisons:** Figure 6 shows the enhanced results from those recent researches and our RetinexGAN. The RUAS-dark Liu et al. (2021) can induce pronounced overexposure problems when engaged in the processing of low light image enhancement, subsequently culminating in the diminishment of feature details. From the enhanced results in the third row we can find that the enhancement effect of some unsupervised methods like EnlightenGAN Jiang et al. (2021) and Zero-DCE++ Li et al. (2021) in crucial low-light regions is not pronounced, and the restoration of pedestrian facial attributes remains insufficient. Unfortunately, the SCI-difficult Ma et al. (2022) method produces color deviation when restoring low light images. Regarding the DeepLPF Moran et al. (2020) scheme, although it demonstrates commendable performance in handling particular low-light images, its generalization capacity is limited due to its inability to effectively recover textural details from darker regions.

Table 1: Quantitative Comparison With State-of-the-Arts on the NPE, MEF, VV Datasets. (The best result is in red whereas the second best one is in blue under each case.)

| Datasets | LOL | | | LSRW | | | MEF | | |
| --- | --- | --- | --- | --- | --- | --- | --- | --- | --- |
| | Paired ✔ | UnPaired ✗ | | Paired ✔ | UnPaired ✗ | | Paired ✗ | UnPaired ✔ | |
| Metrics | PSNR↑ | SSIM↑ | NIQE↓ | PSNR↑ | SSIM↑ | NIQE↓ | CEIQ↑ | PIQE↓ | NIQE↓ |
| SCI (Ma et al. (2022)) | 18.41 | 0.682 | 7.35 | 13.82 | 0.635 | 8.88 | 4.06 | 29.736 | 7.63 |
| LIME (Guo et al. (2016)) | 17.22 | 0.724 | 6.24 | 14.78 | 0.661 | 6.12 | 3.66 | 30.720 | 6.19 |
| RetinexNet (Chen Wei (2018)) | 15.99 | 0.718 | 6.08 | 14.37 | 0.665 | 7.86 | 3.75 | 30.738 | 5.82 |
| RUAS (Liu et al. (2021)) | 12.08 | 0.651 | 8.93 | 11.20 | 0.620 | 10.64 | 2.32 | 33.820 | 10.08 |
| KinD++ (Zhang et al. (2021)) | 9.31 | 0.565 | 11.25 | 8.91 | 0.552 | 10.53 | 2.11 | 38.351 | 9.96 |
| EnlightenGAN (Jiang et al. (2021)) | 19.13 | 0.745 | 5.86 | 15.89 | 0.694 | 6.33 | 4.33 | 29.756 | 4.41 |
| Zero-DCE++ (Li et al. (2021)) | 18.67 | 0.740 | 5.44 | 16.05 | 0.706 | 6.15 | 4.22 | 29.747 | 4.67 |
| DeepLPF (Moran et al. (2020)) | 10.35 | 0.582 | 9.02 | 9.98 | 0.554 | 10.02 | 1.77 | 36.576 | 8.25 |
| RetinexGAN | 18.82 | 0.763 | 4.98 | 16.57 | 0.705 | 6.08 | 4.45 | 27.768 | 4.35 |

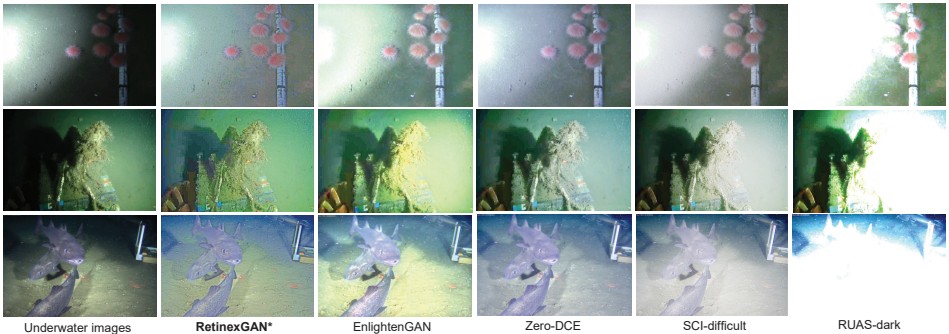

Underwater images    **RetinexGAN\***    EnlightenGAN    Zero-DCE    SCI-difficult    RUAS-dark

Figure 8: Application results in underwater image enhancement.

Another visualization results with respect to VV dataset are shown in Figure 7. We present test results for more methods in Figure 7. The RUAS approaches Liu et al. (2021) (RUAS-dark, RUAS-lol) have the same problem as it shows in Figure 6. The detailed features in the background are blurry due to excessive exposure. Compared to the RUAS-dark, RUAS-lol and RUAS upe Liu et al. (2021) exhibits completely opposite performance. The EnlightenGAN Jiang et al. (2021) and Zero-DCE++ Li et al. (2021) have similar performance. The brightness of regions highlighted by red bounding boxes are significantly insufficient. The performance of SCI-difficult is relatively better than that of SCI-medium and SCI-easy Ma et al. (2022). However, by observing the highlighted regions in the red bounding boxes, it can be observed that the brightness and saturation of its colors are not as good as those of RetinexGAN. The enhanced result from traditional LIME Guo et al. (2016) method also illustrate that there exists some regions with color distortion and artifacts in it. Supervised learning based methods like DeepLPF Moran et al. (2020) and KinD++ Zhang et al. (2021) all fail to restore images with normal-exposure from dark areas. The RetinexNet performs, on the whole, much better

than DeepLPF Moran et al. (2020) and KinD++Zhang et al. (2021). But we still can find some obvious black edges on the edges of objects in the enhanced image through observing highlighted regions. More testing results on other datasets can be found in Appendix (Figure 10-Figure 13).

**Quantitative Comparisons:** In the following, we quantitatively compare our proposed RetinexGAN with various state-of-the-art enhancement methods using image evaluation metrics. For full-reference image quality assessment, we employ the Peak Signal-to-Noise Ratio (PSNR, dB), Structural Similarity (SSIM), Natural Image Quality Evaluator (NIQE), contrast enhancement based contrast-changed image quality measure (CEIQ) Yan et al. (2019) and Perception-based Image Quality Evaluator (PIQE) Venkatanath et al. (2015) to quantitatively compare the performance of different methods on those datasets participated in our experiments. Part of the results are

Table 2: mAP for Face Detection in the Dark Under Different IoU Thresholds (The best result is in red whereas the second best one is in blue under each case.)

| Methods | IoU thresholds | | |
|---|---|---|---|
| | 0.5 | 0.7 | 0.9 |
| SCI Ma et al. (2022) | 0.1925 | 0.0682 | 0.00134 |
| LIME Guo et al. (2016) | 0.1768 | 0.0544 | 0.00108 |
| RetinexNet Chen Wei (2018) | 0.1962 | 0.0623 | 0.00131 |
| RUAS Liu et al. (2021) | 0.1722 | 0.0620 | 0.00129 |
| KinD++ Zhang et al. (2021) | 0.1138 | 0.0324 | 0.00091 |
| EnlightenGAN Jiang et al. (2021) | 0.2075 | 0.0724 | 0.00165 |
| Zero-DCE++ Li et al. (2021) | 0.2191 | 0.0783 | 0.00174 |
| DeepLPF Moran et al. (2020) | 0.1556 | 0.0521 | 0.00107 |
| RetinexGAN | 0.2183 | 0.0813 | 0.00176 |

shown in Table 1 and Table 3 (see it in Appendix). It is obvious that our results are most favored by the subjects for the NPE Wang et al. (2013), MEF Ma et al. (2015), VV datasets. Moreover, our proposed RetinexGAN has the second best score in IT compared to those state-of-the-art methods. Hence, our method can commendably navigate the balance between performance and efficiency of the model that means our RetinexGAN can ensure high-quality enhancement effects while improving model efficiency. The evaluation metrics on other datasets please refer to Appendix.

**Application in Dark Face Detection and UIE:** We analyze the effectiveness of RetinexGAN in the context of the face detection task under conditions of reduced illumination. The DARK FACE dataset Yang et al. (2016) that consists of 10000 images captured in the dark environment is used in this experiment. We apply our RetinexGAN as well as other enhancement algorithms to enhance 1000 images from this dataset. Subsequently, Retinaface Deng et al. (2020) is employed to as the baseline model for conducting face detection on the enhanced images. We calculate the mean average precision (mAP) for face detection under various methods through the evaluation tool provided in DARK FACE dataset and list them in Table 2. The results show that the mAP of Retinaface increases considerably compared to that using raw images without enhancement. Meanwhile the RetinexGAN can obtain the highest or second highest mAP scores. The application results of UIE are shown in Figure 8. The underwater images are selected from the OceanDark datasetMarques & Albu (2020); Porto Marques et al. (2019). It is obvious that the results from our proposed Retinex-GAN have high clarity and reasonable exposure level, while the color saturation is the fullest compared to other methods. More visualization results please refer to Appendix.

### 3.4 CONCLUSION

We propose a semi-supervised image enhancement scheme which can be divided into two phase. The first phase is the learning the mapping rules of data-driven Retinex decomposition by devising a set of learning constraints. The second phase is illumination brightening through FPN with a flexible pretrained backbone optimized by a series of unsupervised training loss terms. Extensive experiments demonstrate that our method can commendably navigate the balance between performance and efficiency compared to existing light enhancement methods. In the future work, we will try to seek data-driven Retinex decomposition without the need for paired images, achieving complete unsupervised enhancement while ensuring the efficiency and performance.

### REPRODUCIBILITY STATEMENT

We are in the process of organizing the source code for this research, and all the training data (testing data) and source code will be released. The dataset's partitioning and utilization have been intro-

duced at the beginning of the experimental section, and the image processing program in MATLAB will be released alongside the source code.

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

# A APPENDIX

## A.1 RELATED WORKS

**Traditional Methods:** For databases with fewer images, Earlier methods that relied on traditional methods like Retinex theory Land & McCann (1971); Jang et al. (2012); Li et al. (2018); Gilboa et al. (2004); Bettahar et al. (2011) can significantly improve the visibility of the low-light images and images with color distortion. In Palma Amestoy et al. (2008), motivated by the basic phenomenology of color perception, the authors propose a variational formulation of color contrast enhancement. In Provenzi et al. (2008), a random spray Retinex based white-patch algorithm and gray-world algorithms are combined for establishing a more robust and better performing lightening model. However, these methods will be more time-consuming to process each image, so there is no way to achieve the enhancement for large-scale images in batches quickly.

**Supervised Methods:** Driven by the development of convolution neural network (CNN), some researchers have developed some CNN-based image enhancement algorithms. The initial low-light image enhancement algorithms predominantly relied on supervised deep image enhancement networks, necessitating the availability of paired images. For example, in Lore et al. (2017), a deep auto-encoder based approach is proposed to extract and denoise the signal features from low-light images and improve contrast. InChen et al. (2018), the authors propose a deep learning-based method for enhancing low-light images by leveraging raw sensor data, addressing challenging low-light conditions, and achieving visually appealing results through an end-to-end process. In Tang et al. (2022), the images are divided into content component and attribute component. On this basis, two ResBlocks based encoders are designed to enhance the low-light images. However, as mentioned earlier, the data requirements of such supervised algorithms are stringent, as obtaining paired image datasets can be exceedingly challenging. Furthermore, the scarcity of accessible low-light/normal paired images poses a challenge when relying exclusively on supervised algorithms, as it may lead to overfitting problems and subsequently reduce the network's ability to generalize effectively.

**Unsupervised Methods:** As a response to this challenge, a myriad of unsupervised image enhancement algorithms have emerged, with the primary objective of mitigating the data prerequisites for training. In Jiang et al. (2021), the authors introduce an unsupervised image enhancement method named EnlightenGAN, which utilizes a U-Net generator with self-attentive guidance for automatic adjustment. In Guo et al. (2020); Li et al. (2021), the authors introduce image enhancement methods (Zero-DCE and Zero-DCE++) employing lightweight deep curve estimation networks, resulting in significantly improved algorithm efficiency when compared to EnlightenGAN. In a recent study, the authors in Liu et al. (2021) proposed an unrolling framework inspired by Retinex, which incorporates architecture search techniques. And in Ma et al. (2022), the authors present a self-calibrated illumination learning framework, designed to achieve efficient, adaptable, and resilient low-light image enhancement. Those unsupervised algorithms have, to a certain extent, reduced network size and the number of trainable parameters, thereby enhancing efficiency. However, their enhancement capability is remarkably limited when facing images with extremely low illumination. We show some corresponding results in Fig.1. While supervised learning-based methods can yield enhanced images with superior visual quality, they often incur higher Floating Point Operations (FLOPs) and longer inference times. Some unsupervised methods like EnlightenGAN also have large size of trainable parameters to be optimized that reduces training efficiency. Accordingly, how to navigate the balance between performance and efficiency of unsupervised image enhancement methods is still a critical issue that urgently needs to be addressed.

## A.2 ARCHITECTURE OF DEEP DECOMPOSITION NETWORK

The detailed architecture of deep decomposition network is shown in Fig.2. It is made up of six modules, including one convolution layer, four ResBlocks and one activation layer. The first convolution layer calculates 16 feature maps with $7 \times 7$ kernels. The ResBlock is constructed by connecting ConnectUnit and ResidualUnit in series. The ConnectUnit in the first ResBlock consists of a convolution layer that can calculate 32 feature maps with $3 \times 3$ kernels. The ResidualUnit in the first ResBlock consists of two convolution layer that also can calculate 32 feature maps with $3 \times 3$ kernels. All the convolution layers in the second ResBlock can calculate 64 feature maps with $3 \times 3$ kernels. Analogously, the convolution layers in the third ResBlock can calculate 128 feature maps with $3 \times 3$

kernels. The convolution layers in the last ResBlock calculate 6 feature maps with $3 \times 3$ kernels. We use LeakyReLu to replace the conventional ReLu activation function in those four ResBlocks for avoiding vanishing gradient problem. The Sigmoid activation layer is adopted finally to avoid data overflow.

### A.3 Architecture of Global Discriminator

The global discriminator is constructed for distinguishing synthetic images from real normal exposure images. It consists of four ResBlocks and final output can be obtained through this four ResBlocks and fully connected layers. We replace the sigmoid function with the least-square GAN Mao et al. (2017) and the training loss of discriminator is:

$$J_D = \mathbb{E}_{x_r \sim \mathbb{P}_{\text{real}}} \left( D\left(x_r\right) - 1 \right)^2 + \mathbb{E}_{x_f \sim \mathbb{P}_{\text{fake}}} \left( D\left(x_f\right) \right)^2 \tag{14}$$

where $\mathbb{P}_{\text{real}}$ stand for the distribution of the normal-exposure images. $\mathbb{P}_{\text{fake}}$ is the distribution of the synthetic images using the adjusted illumination from the generator.

### A.4 Experiment Results

In the experiments of our research, we use synthetic images and real-world images to verify the effectiveness of proposed RetinexGAN. And some additional experimental results which can not be illustrated in the main paper due to the length limitation are shown in this section. Firstly, we provide the pseudo codes of the training procedure for making our algorithm more transparent and easier for readers to follow. Then we show some additional results in ablation study with respect to our designed loss terms. Finally, sufficient benchmark evaluation results are illustrated to supplement the experimental results in the main paper.

#### A.4.1 Pseudo Codes of the RetinexGAN Training

---
**Algorithm 1:** RetinexGAN Training

---
**Input:** Paired Training dataset $X$, total number of paired training samples $N$, training steps $N_{ep}$, positive integer $d_{iter}$, batch size $n_b$, unpaired training dataset $X_u$

**Output:** Image enhancement model $\mathcal{M}$

1   calculate the number of times a sample needs to be traversed in each training step
     $n_{bs} = N//n_b$;

2   **for** $i = 1; i \leq N_{ep}; i + + $ **do**

3      **for** $j = 1; j \leq n_{bs}; j + + $ **do**

4          **if** $j//d_{iter} \neq 0$ **then**

5              Get a batch of training samples $x$ from paired image dataset $X$;

6              Calculate the Retinex decomposition loss
             $\mathcal{L}\left(x, \theta\right) \leftarrow \mathcal{W}_r \mathcal{L}_r + \mathcal{W}_{do} \mathcal{L}_{do} + \mathcal{W}_c \mathcal{L}_c + \mathcal{W}_f \mathcal{L}_f + \mathcal{W}_{tv} \mathcal{L}_{tv}$;

7              Update the trainable parameters $\theta$ of the deep Retinex decomposition network using *Adam* optimizer $\theta \leftarrow \text{Adam}\left(\mathcal{L}\left(x, \theta\right)\right)$;

8              **Continue**;

9          **else**

10              Get a batch of training samples $x_u$ from unpaired image dataset $X_u$;

11              Calculate the non-reference illumination brightening loss
             $\mathcal{J}\left(x_u, \eta\right) \leftarrow \mathcal{O}_b \mathcal{J}_b + \mathcal{O}_f \mathcal{J}_f + \mathcal{O}_a \mathcal{J}_a + \mathcal{O}_{adv} \mathcal{J}_{adv} + \mathcal{L}$;

12              Update the trainable parameters $\eta$ of the illumination brightening network using *Adam* optimizer $\eta \leftarrow \text{Adam}\left(\mathcal{J}\left(x_u, \eta\right)\right)$;

13              Calculate the global discriminator loss of the LSGAN and update the parameters of the discriminator using *Adam* optimizer.

14   **Return** Image enhancement model $\mathcal{M}\left(\theta, \eta\right)$;

---

### A.4.2 ABLATION STUDY

The images in Figure 9 show the results of ablation study with respect to each loss term. And they clearly demonstrate the importance of them in improving the performance of our model in illumination brightening.

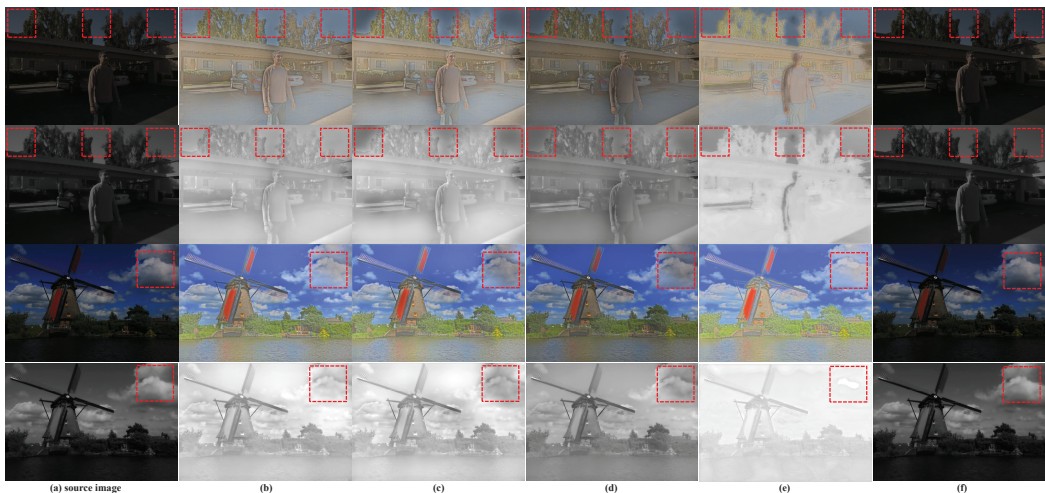

Figure 9: Visual comparison from the ablation study (contribution of losses) of RetinexGAN. Row 1 (Column (b)-Column (f)) display the final enhanced results. Row 2 (Column (b)-Column (f)) are the adjusted illumination maps generated by illumination generator. Column (a) shows the source image (Row 1) and its illumination maps (Row 2). (b) RetinexGAN. (c) $w/o \mathcal{J}_a$. (d) $w/o \mathcal{J}_{adv}$. (e) $w/o \mathcal{J}_f$. (f) $w/o \mathcal{J}_b$

### A.4.3 BENCHMARK EVALUATION RESULTS

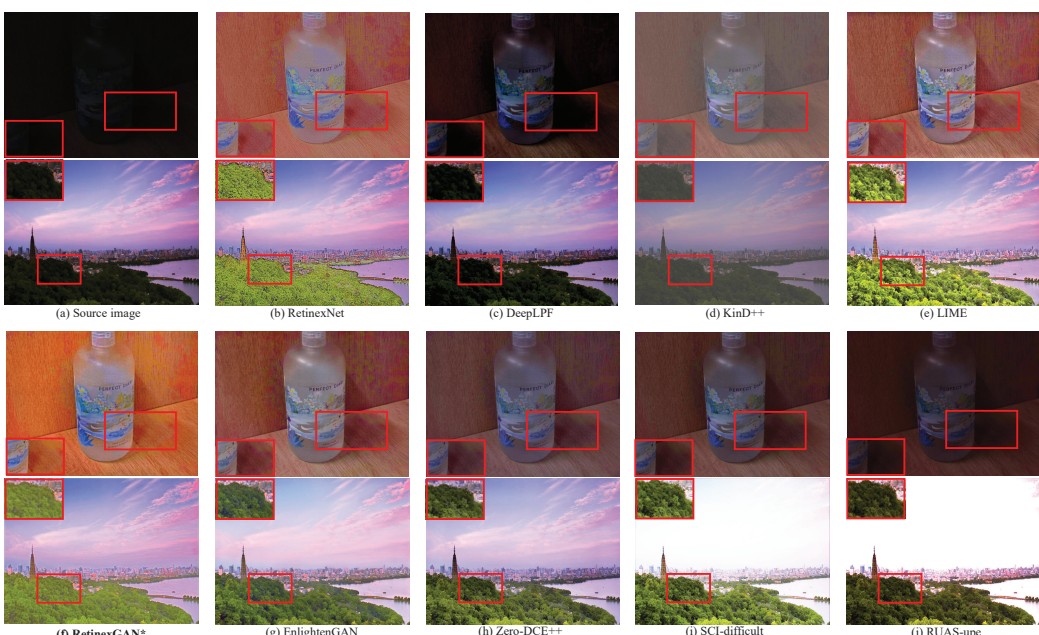

Figure 10: Enhancement Results from LSRW dataset and NPE dataset

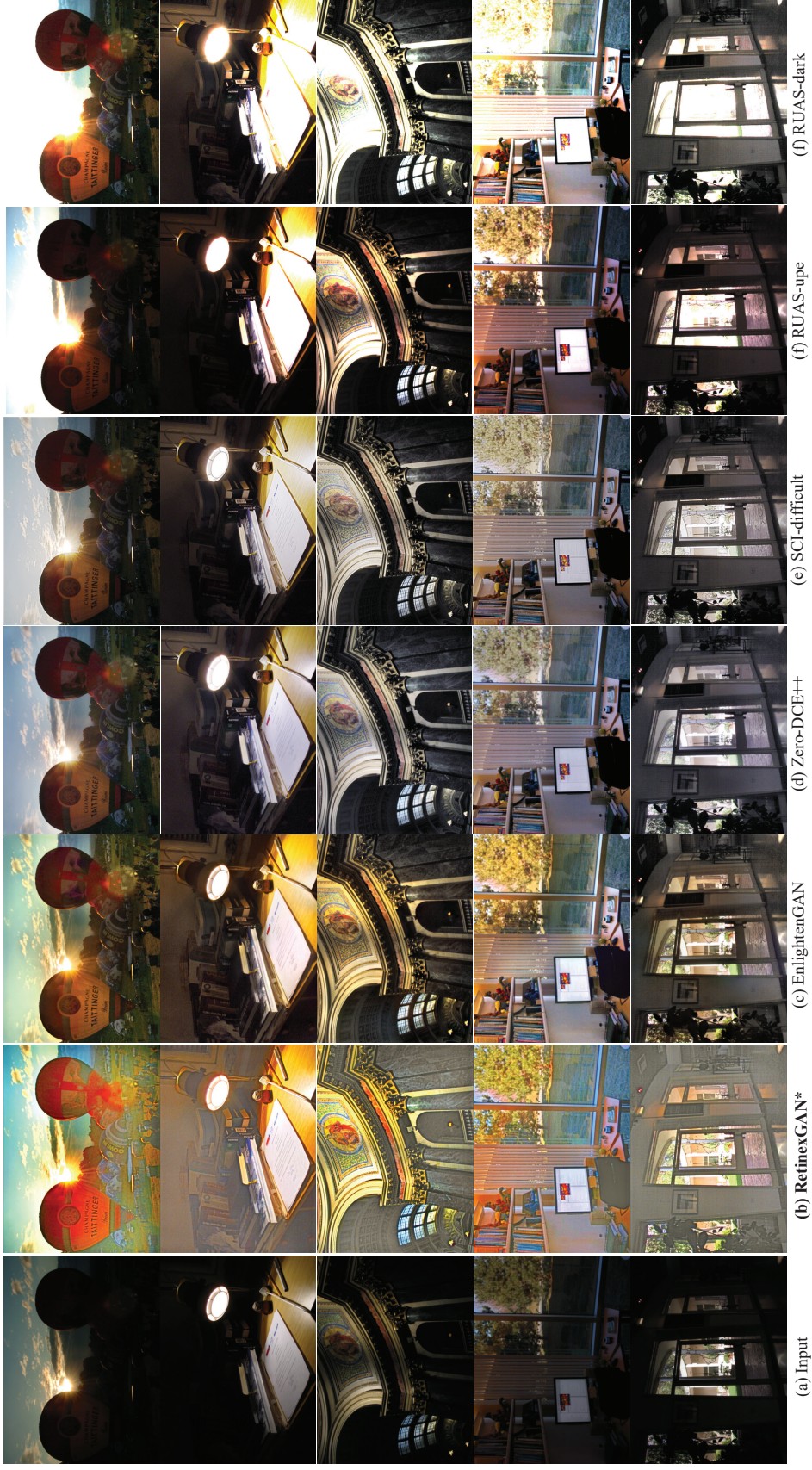

Figure 11: Enhancement Results from MEF dataset

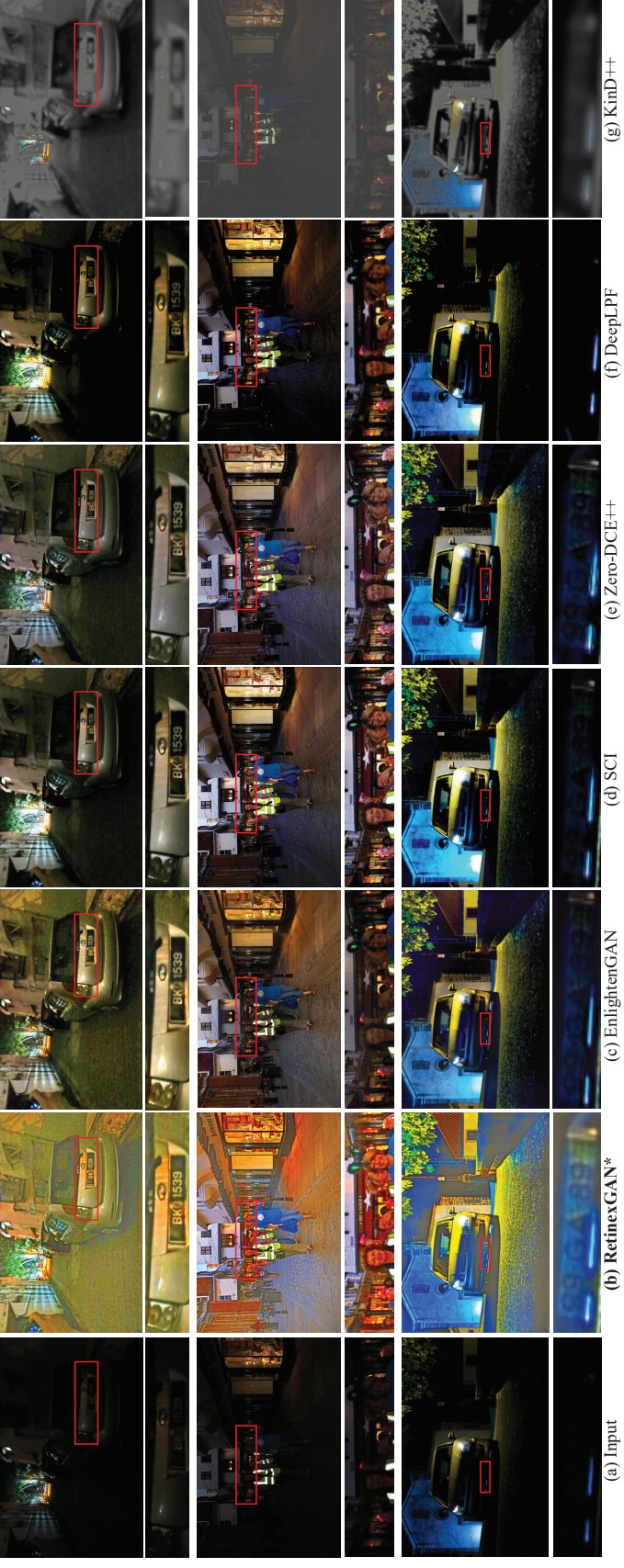

Figure 12: Enhancement Results from real-world captured ExDark dataset

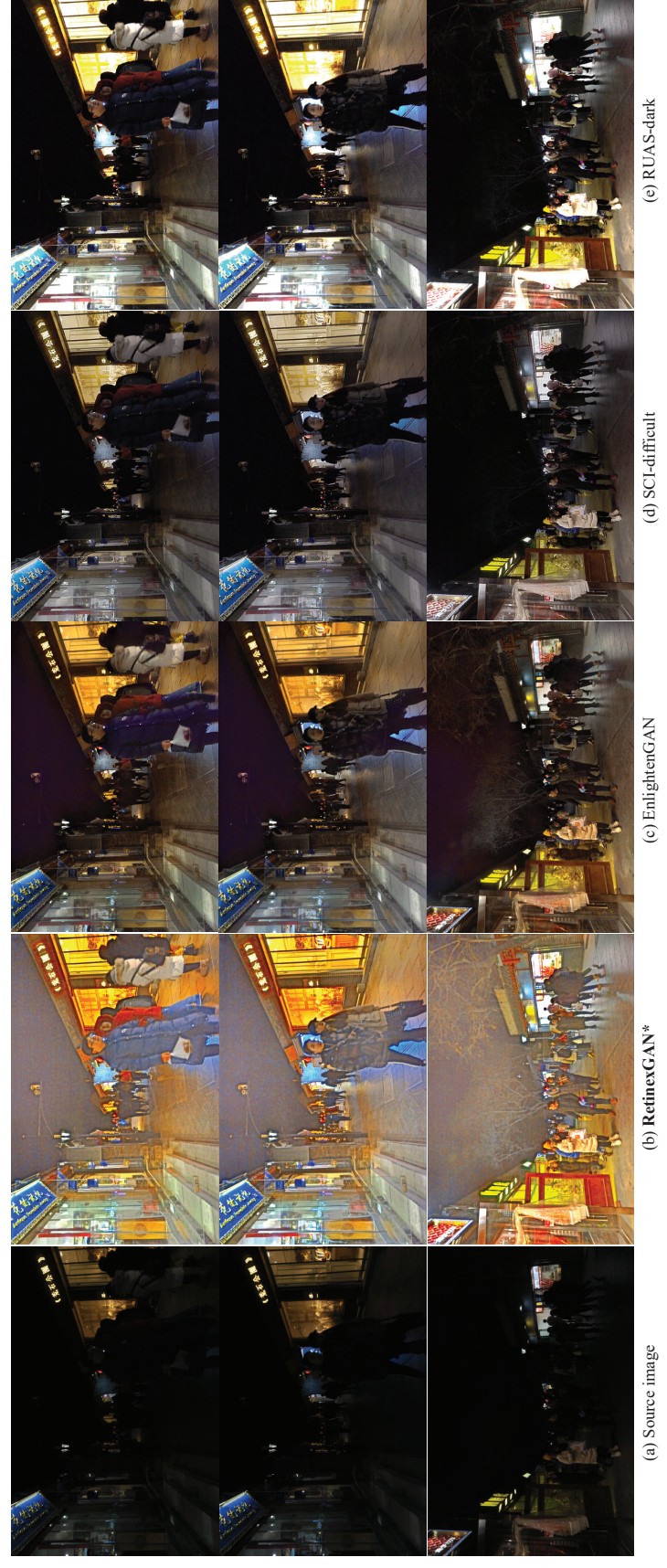

Figure 13: Enhancement Results from DarkFace dataset

Table 3: Quantitative Comparison With State-of-the-Arts on the OceanDark, VV, ExDark Datasets. (The best result is in red whereas the second best one is in blue under each case.)

| Dataset | ExDark | | | VV | | | OceanDark | | | Efficiency | | |
|---|---|---|---|---|---|---|---|---|---|---|---|---|
| Algorithms | CEIQ↑ | NIQE↓ | PIQE↓ | CEIQ↑ | NIQE↓ | PIQE↓ | CEIQ↑ | NIQE↓ | PIQE↓ | IT(s)[1] | Pa(M)[2] | FLOPs(G) |
| SCI | 3.02 | 6.25 | 33.79 | 3.21 | 5.89 | 32.91 | 2.68 | 5.99 | 33.01 | 0.1777 | 0.0003 | 0.1645 |
| LIME | 3.43 | 7.54 | 32.77 | 3.36 | 7.73 | 32.82 | 2.97 | 6.12 | 32.82 | 0.6548 | / | / |
| RetinexNet | 2.87 | 7.67 | 35.76 | 3.09 | 7.92 | 34.29 | 2.46 | 7.69 | 34.65 | 0.1524 | 0.56 | 535.34 |
| RUAS | 1.98 | 8.45 | 39.26 | 1.99 | 8.01 | 38.37 | 1.82 | 8.22 | 38.78 | 0.1835 | 0.0034 | 1.00 |
| KinD++ | 1.33 | 10.17 | 40.98 | 1.22 | 10.25 | 40.11 | 1.15 | 9.98 | 41.89 | 9.97 | 8.27 | 198.67 |
| EnlightenGAN | 3.23 | 5.88 | 33.47 | 3.45 | 5.68 | 32.98 | 2.90 | 6.15 | 32.46 | 0.0987 | 8.63 | 203.77 |
| Zero-DCE++ | 3.01 | 5.23 | 32.96 | 3.23 | 5.79 | 33.02 | 2.96 | 6.02 | 33.18 | 0.0033 | 0.01 | 0.05 |
| DeepLPF | 1.06 | 10.23 | 41.25 | 1.02 | 9.88 | 42.37 | 1.02 | 12.58 | 44.87 | 0.1256 | 1.71 | 19.35 |
| RetinexGAN | 3.58 | 5.66 | 31.58 | 3.94 | 5.73 | 31.24 | 2.98 | 5.93 | 32.02 | 0.0478 | 1.18 | 49.46 |

[1] IT is the abbreviation for inference time
[2] Pa is the abbreviation for total number of trainable parameters

Table 4: Influence of Different Backbones on Model Efficiency and Performance (The best result is in red.)

| | PSNR↑ | NIQE↓ | FLOPs(G) | Pa(M)[1] | IT(s)[2] |
|---|---|---|---|---|---|
| **MobileNet** | 23.45 | 4.22 | 49.46 | 1.18 | 0.0478 |
| SENet | 22.83 | 4.53 | 229.98 | 76.10 | 0.4208 |
| DenseNet | 23.04 | 4.25 | 134.16 | 10.21 | 0.1635 |
| Inception[3] | 23.88 | 3.95 | 249.63 | 5.19 | 0.2192 |

[1] Pa is the abbreviation for total number of trainable parameters
[2] IT is the abbreviation for inference time
[3] Inception is the abbreviation for **Inception-ResNet-v2**

