# OpenReview forum: "RetinexGAN Enables More Robust Low-Light Image Enhancement Via Retinex Decomposition Based Unsupervised Illumination Brightening"
_ICLR.cc/2024/Conference — ICLR 2024 Conference Withdrawn Submission_

### Official Review · Reviewer_p4zt · 2023-10-30

**Soundness:** 3 good
**Presentation:** 3 good
**Contribution:** 2 fair
**Rating:** 3
**Confidence:** 4

**Summary:**

This paper proposed a Retinex-decomposition-based generative adversarial network for low-light image enhancement, which is trained with unpaired samples. The experiments showed the superior performance of the proposed neural network over some recent approaches.

**Strengths:**

+ Combination of GAN, Retinex model, and unpaired learning.
+ Interpretability of the neural network architecture from due to the use Retinex model.

**Weaknesses:**

- The compared methods in the expriements are somehow out of data. Only one compared method is published at or after 2022. This makes the experiemtns not convincing.
- GAN, unpaired training, and Reintex model are widely studied and utillized in low-light enhancement. The papers lacks a detailed discussion with existing methods and ideed it seems that the proposed components have no big differences from the existing ones.

**Questions:**

See the Weakeness part.

---

> ### Author Response · Authors · 2023-11-20
> **Response to Reviewer p4zt**
>
> We sincerely thank the reviewer for carefully reading our paper and pointing out our shortcomings. We have carefully incorporated them in the revised paper. In the following, your comments are first stated and then followed by our point-by-point responses.
>
> [Q1] The compared methods in the expriements are somehow out of data. Only one compared method is published at or after 2022. This makes the experiemtns not convincing.
>
> [A1] Thank you for pointing this problem. According to your suggestions, we have provided more qualitative results on the LOL dataset and except for that, we also show some visualization results on the LOL dataset that contains significant noise.More importantly, we include comparisons with many methods proposed in 2023 such as NeRCo (ICCV2023), CLIP-LIT (ICCV2023) and Neural Preset(CVPR2023).
>
> [Q2] GAN, unpaired training, and Reintex model are widely studied and utillized in low-light enhancement. The papers lacks a detailed discussion with existing methods and ideed it seems that the proposed components have no big differences from the existing ones.
>
> [A2] Thank you for pointing out this concern. There is a significant difference between the FPN illumination generator and self-attention mechanism in our work and other works. For self-attention mechanism, we first used the value channel in the HSV channel model and used exponential average pooling to obtain the self attention map, which allows the model to focus more on modifying the darker areas in the map and enable it to be applied in more flexible and diverse scenarios. For FPN network, we have adopted the basic architecture of FPN network and applied it to adjust the illumination map of low-light images for the first time. Specifically, we have designed a switchable pretrained backbones for the downsampling part of the FPN based illumination map adjustment network. Users can choose between the timeliness and performance of the model according to their actual needs. At the same time, in order to accelerate model convergence, we have designed a self-attention weighted global residual connection for the FPN network. These designs make our model very different from other models, That is where the innovation of this study lies.

---

### Official Review · Reviewer_TqR9 · 2023-10-30

**Soundness:** 2 fair
**Presentation:** 2 fair
**Contribution:** 1 poor
**Rating:** 3
**Confidence:** 5

**Summary:**

This paper presents a semi-supervised illumination brightening framework for low-light image enhancement. A lightweight CNN model is used to achieve Retinex decomposition, and then a feature pyramid network is employed to brighten the illumination maps in a unsupervised manner. The proposed method has been evaluated on several datasets, and improved results have been achieved.

**Strengths:**

* The paper is easy to follow.
* The proposed framework achieves improved performances on both synthetic and real-world low light images.

**Weaknesses:**

* My main concern about this paper is its limited novelty and technical contribution. The adopted FPN network and the spatial attention mechanism are very common strategies in many research fields.
* For the experimental results, although improved qualitative results have been obtained, the visual results in Fig.1(a), Fig.6 and Fig.7 still look poor. For better comparisons, can the authors provide more comparison results with ground truth as a reference? For downstream applications, it may be more intuitive to use P-R curves for comparisons (as Zero-DCE).
* The paper’s readability is poor. A lot of technical details are missing. The writing should be improved for a better review.
* The literature survey should be more elaborate. Several newly published works are not discussed, especially the supervised methods.

**Questions:**

Other issue:
* The abbreviation "RetinexGAN" has been used in "Ma et al., RetinexGAN: Unsupervised low-light enhancement with two-layer convolutional decomposition networks, IEEE Access".

---

> ### Author Response · Authors · 2023-11-20
> **Response to Reviewer TqR9**
>
> We sincerely thank the reviewer for carefully reading our paper and pointing out our shortcomings. We have carefully incorporated them in the revised paper. In the following, your comments are first stated and then followed by our point-by-point responses.
>
> [Q1] My main concern about this paper is its limited novelty and technical contribution. The adopted FPN network and the spatial attention mechanism are very common strategies in many research fields.
>
> [A1] Thank you for pointing out this concern. There is a significant difference between the FPN illumination generator and self-attention mechanism in our work and other works. For self-attention mechanism, we first used the value channel in the HSV channel model and used exponential average pooling to obtain the self attention map, which allows the model to focus more on modifying the darker areas in the map and enable it to be applied in more flexible and diverse scenarios. For FPN network, we have adopted the basic architecture of FPN network and applied it to adjust the illumination map of low-light images for the first time. Specifically, we have designed a switchable pretrained backbones for the downsampling part of the FPN based illumination map adjustment network. Users can choose between the timeliness and performance of the model according to their actual needs. At the same time, in order to accelerate model convergence, we have designed a self-attention weighted global residual connection for the FPN network. These designs make our model very different from other models, That is where the innovation of this study lies.
>
> [Q2] For the experimental results, although improved qualitative results have been obtained, the visual results in Fig.1(a), Fig.6 and Fig.7 still look poor. For better comparisons, can the authors provide more comparison results with ground truth as a reference? For downstream applications, it may be more intuitive to use P-R curves for comparisons (as Zero-DCE).
>
> [A2] Thank you for pointing out this concern. Since we did not optimize the model to the best during hyperparameter tuning, the enhanced version of low-light images exhibit evident color shifts and artifacts. We have readjusted the hyperparameters of our enhancement model to improve the performance and eliminate color deviation and artifacts in the enhanced images. According to your suggestions, we have provided more comparison results with ground truth as a reference and use P-R curves to illustrate the application in dark face detection in the revised manuscript.
>
> [Q3]The paper’s readability is poor. A lot of technical details are missing. The writing should be improved for a better review.
>
> [A3] Thank you for pointing out this concern. We have improved the writing of this paper and improved its readability.
>
> [Q4]The literature survey should be more elaborate. Several newly published works are not discussed, especially the supervised methods.
>
> [A4] Thank you for pointing out this concern. We have revised the writing of literature survey of this paper and disscussed more supervised methods.
>
> [Q5]The abbreviation "RetinexGAN" has been used in "Ma et al., RetinexGAN: Unsupervised low-light enhancement with two-layer convolutional decomposition networks, IEEE Access".
>
> [A5] Thank you for pointing out this problem. We have renamed our method.

---

### Official Review · Reviewer_tFmg · 2023-10-31

**Soundness:** 1 poor
**Presentation:** 2 fair
**Contribution:** 1 poor
**Rating:** 1
**Confidence:** 5

**Summary:**

The work introduces a decomposition network based on the Retinex theory. Subsequently, it incorporates a Generative Adversarial Learning strategy by constructing an illumination generation network based on FPN to further enhance low-light images captured in unknown scenarios. A series of related experiments demonstrate the superiority of the proposed method and the effectiveness of the various modules and constraints designed.

**Strengths:**

The approach of building an illumination enhancement network by introducing FPN is quite intriguing. The overall framework's process and structural details are described clearly, and the overall presentation is commendable.

**Weaknesses:**

1. Unclear Motivation: The motivations behind the various components in the overall framework constructed by the author seem to lack clear explanations. For instance, the initial intention of decomposition followed by enhancement, the benefits of introducing FPN, and the reasons for employing different training strategies at different stages, among other aspects, are not adequately addressed. Unfortunately, the author appears to have not provided relevant analyses for these issues. Furthermore, it is suggested that the author should provide additional evidence in their response to support their claims.
2. Lack of Novelty: Whether it's the decomposition network based on Retinex, generative adversarial networks, attention mechanisms, or the range of introduced loss functions, these are common approaches for addressing low-light image enhancement problems. Additionally, while the idea of introducing FPN is promising, the lack of a clear explanation by the author impacts the inevitability of the contribution's novelty. In other words, this work appears more like a combination of existing effective techniques, leaving room for increased novelty.
3. Unconvincing Experimental Results: Most of the visual results presented in the manuscript exhibit evident color shifts, and some even contain artifacts. In comparison to other methods, the superiority of the proposed approach in terms of qualitative results is hard to establish.
4. Need for Improvement in Experimental Settings and Presentation: From the manuscript, it can be seen that the author conducted a series of comparative experiments on the LOL dataset, but it seems that no qualitative results are provided. I am curious about the performance of the proposed method on data from the LOL dataset that contains significant noise. Furthermore, it appears that no comparisons were made with low-light image enhancement methods proposed in 2023. As far as I am aware, there have been many recent representative advances in this field, and it is hoped that the author can include comparisons with them to more comprehensively validate the effectiveness of the proposed method.

Overall, this work requires significant improvement in several aspects. The ablation experiments section also suffers from a lack of comprehensiveness. It is hoped that the author can provide sufficiently detailed responses to the above-mentioned issues.

**Questions:**

Please refer to Weaknesses.

---

> ### Author Response · Authors · 2023-11-20
> **Response to Reviewer tFmg**
>
> We sincerely thank the reviewer for the time reading our paper. We have carefully incorporated them in the revised paper. In the following, your comments are first stated and then followed by our point-by-point responses.
>
> [Q1] Unclear Motivation: The motivations behind the various components in the overall framework constructed by the author seem to lack clear explanations. For instance, the initial intention of decomposition followed by enhancement, the benefits of introducing FPN, and the reasons for employing different training strategies at different stages, among other aspects, are not adequately addressed. Unfortunately, the author appears to have not provided relevant analyses for these issues. Furthermore, it is suggested that the author should provide additional evidence in their response to support their claims.
>
> [A1] Thank you for pointing out these concerns. First, the initial intention of decomposition is to remove the influence of environmental illumination on imaging, obtain reflectance components that are not dependent on the external environment. Then, by adjusting the illumination maps and recombining the illumination and reflectance maps, the enhanced image is obtained. Second, through feature propagation and fusion, FPN can combine high-level semantic information with low-level detail information, enabling the network to simultaneously obtain rich semantic and detail information during illuminance map restoration, thereby improving the robustness and generalization ability of illuminance map restoration. We provide algorithm pseudocode in the appendix to demonstrate the training logic of the model in detail. According to your suggestions, we have provided relevant analyses in the revised manuscript.
>
> [Q2] Lack of Novelty: Whether it's the decomposition network based on Retinex, generative adversarial networks, attention mechanisms, or the range of introduced loss functions, these are common approaches for addressing low-light image enhancement problems. Additionally, while the idea of introducing FPN is promising, the lack of a clear explanation by the author impacts the inevitability of the contribution's novelty. In other words, this work appears more like a combination of existing effective techniques, leaving room for increased novelty.
>
> [A2] Actually, our research is certainly not a combination of existing effective techniques. You mention that Retinex theory, generative adversarial networks, attention mechanisms and even designed loss functions are common. However, our data-driven Retinex decomposition is realized through some newly proposed learning constraints in section 2.2. There is a significant difference between the self-attention map calculation in our work and other works. We first used the value channel in the HSV channel model and used exponential average pooling to obtain the self attention map, which allows the model to focus more on modifying the darker areas in the map and enable it to be applied in more flexible and diverse scenarios. Accordingly, we specially use the cosine similarity function to design a new loss term (attention loss) for FPN and we have conducted an ablation study on it.
>
> [Q3] Unconvincing Experimental Results: Most of the visual results presented in the manuscript exhibit evident color shifts, and some even contain artifacts. In comparison to other methods, the superiority of the proposed approach in terms of qualitative results is hard to establish.
>
> [A3] Thank you a lot for pointing this problem. Since we did not optimize the model to the best during hyperparameter tuning, the enhanced version of low-light images exhibit evident color shifts and artifacts. We have readjusted the hyperparameters of our enhancement model to improve the performance and eliminate color deviation and artifacts in the enhanced images. All experimental parameter corrections are highlighted in blue in the main text.
>
> [Q4] Need for Improvement in Experimental Settings and Presentation: From the manuscript, it can be seen that the author conducted a series of comparative experiments on the LOL dataset, but it seems that no qualitative results are provided. I am curious about the performance of the proposed method on data from the LOL dataset that contains significant noise. Furthermore, it appears that no comparisons were made with low-light image enhancement methods proposed in 2023.
>
> [A4] Thank you for pointing this problem. According to your suggestions, we have provided more qualitative results on the LOL dataset and except for that, we also show some visualization results on the LOL dataset that contains significant noise.More importantly, we include comparisons with many methods proposed in 2023 such as NeRCo (ICCV2023), CLIP-LIT (ICCV2023) and Neural Preset(CVPR2023).

---

### Official Review · Reviewer_6g5x · 2023-10-31

**Soundness:** 2 fair
**Presentation:** 2 fair
**Contribution:** 2 fair
**Rating:** 3
**Confidence:** 5

**Summary:**

The research direction of this article is low-light image enhancement, which mainly solves the image degradation problem through Retinex decomposition theory and FPN-based attention mechanism. The contribution of the article is to simultaneously provide flexibility in low-light image enhancement by leveraging lightweight cellular neural networks for data-driven Retinex decomposition and leveraging FPN with different types of pre-trained backbones for downsampling in illumination generation.

**Strengths:**

1. The main purpose of this article is to propose a flexible framework for enhancing low-light images, which has practical applications.
2. Motivation of this paper is clear and the main idea is easy to understand.

**Weaknesses:**

1. The author did mention that the work aims to find a balance between efficiency and performance, but the main body of the text primarily focuses on the analysis of performance. Furthermore, based on the experimental results regarding computational efficiency in the appendix, it seems that the proposed method may not have achieved a sufficiently prominent advantage in terms of efficiency.
2. In the second section of the method introduction, the author dedicates a significant amount of space to explaining specific network structures and the various constraints introduced, without providing an explanation for the reasons. In other words, the motivations for the various components in the proposed framework may require additional clarification from the author.
3. Compared to other methods, the results shown in Figure 1, Figure 7, and Figure 11 do not appear to be the best. In Figure 5, there are even instances of artifacts. It is suggested that the authors provide an explanation for the phenomena mentioned above.
4. In the ablation experiment section, the author only analyzed the loss function. To further validate the effectiveness of the constructed framework, it is recommended that the author include ablation experiment results for each component of the framework in the manuscript.

**Questions:**

The relevant questions have been raised in the weaknesses section.

---

> ### Author Response · Authors · 2023-11-20
> **Response to Reviewer 6g5x**
>
> Thank you for your constructive comments and suggestions, and they are exceedingly helpful for us to improve our paper. We have carefully incorporated them in the revised paper. In the following, your comments are first stated and then followed by our point-by-point responses.
>
> [Q1] The author did mention that the work aims to find a balance between efficiency and performance, but the main body of the text primarily focuses on the analysis of performance. Furthermore, based on the experimental results regarding computational efficiency in the appendix, it seems that the proposed method may not have achieved a sufficiently prominent advantage in terms of efficiency.
>
> [A1] Thanks very much for pointing out these two important problems of our paper.  First, we show the model performance and model parameter size in Figure 1. And in appendix, we also exhibit the inference time of our model and other state-of-the-art algorithms to show that our work aims to find a balance between efficiency and performance. According to your comments, we have spent more space in the revised manuscript to analyze the advantages of this method in balancing image enhancement efficiency and performance. Second, you said that the proposed method may not have achieved a sufficiently prominent advantage in terms of efficiency. Actually, according to Table 3 in Appendix, our model achieves the second best performance in terms of inference time. However, ZeroDCE++ with the best inference time is significantly weaker than this method in terms of model performance. Our method achieve the best enhancement performance with the second fastest inference time, so we conclude that this method can navigate a balance between performance and efficiency.
>
> [Q2] In the second section of the method introduction, the author dedicates a significant amount of space to explaining specific network structures and the various constraints introduced, without providing an explanation for the reasons. In other words, the motivations for the various components in the proposed framework may require additional clarification from the author.
>
> [A2] Thanks for your constructive comments. We have further elaborated and explained the various components, loss terms and constraints in network training.
>
> [Q3] Compared to other methods, the results shown in Figure 1, Figure 7, and Figure 11 do not appear to be the best. In Figure 5, there are even instances of artifacts. It is suggested that the authors provide an explanation for the phenomena mentioned above.
>
> [A3] Thanks for pointing out this concern. We are very sorry for the presentation of poor enhancement results in the paper due to our failure to optimize the model to the best during hyperparameter tuning. But after adjusting the hyperparameters and network structure parameters, we have achieved the best enhancement effect. All experimental parameter corrections are highlighted in blue in the main text.
>
> [Q4] In the ablation experiment section, the author only analyzed the loss function. To further validate the effectiveness of the constructed framework, it is recommended that the author include ablation experiment results for each component of the framework in the manuscript.
>
> [A4] In the ablation study, we did not only analyze the components of the loss function, but also conducted ablation experiments on the attention mechanism in FPN and the downsampling pretraining backbones in FPN. According to your suggestion, we have conducted supplementary ablation experimental analysis on the number of ResBlocks in the decomposition network.